# CRITIC-GUIDED REINFORCEMENT UNLEARNING IN TEXT-TO-IMAGE DIFFUSION

## ABSTRACT

Machine unlearning in text-to-image diffusion models aims to remove targeted concepts while preserving overall utility. Prior diffusion unlearning methods typically rely on supervised weight edits or global penalties; reinforcement-learning (RL) approaches, while flexible, often optimize sparse end-of-trajectory rewards, yielding high-variance updates and weak credit assignment. We present a general RL framework for diffusion unlearning that treats denoising as a sequential decision process and introduces a timestep-aware critic with noisy-step rewards. Concretely, we train a CLIP-based reward predictor on noisy latents and use its per-step signal to compute advantage estimates for policy-gradient updates of the reverse diffusion kernel. Our algorithm is simple to implement, supports off-policy reuse, and plugs into standard text-to-image backbones. Across multiple concepts, the method achieves better or comparable forgetting to strong baselines while maintaining image quality and benign prompt fidelity; ablations show that (i) per-step critics and (ii) noisy-conditioned rewards are key to stability and effectiveness. We release code and evaluation scripts to facilitate reproducibility and future research on RL-based diffusion unlearning.

## 1 INTRODUCTION

Text-to-image diffusion models (DMs) are the backbone of modern image synthesis, yet real deployments increasingly require unlearning—removing specific concepts (styles, identities, unsafe content) without harming overall utility Ho et al. (2020); Rombach et al. (2022). Existing diffusion unlearning methods typically rely on parameter edits or global penalties and can either over-suppress harmless content or be evaded by adversarial phrasing Gandikota et al. (2023); Gong et al. (2024); Huang et al. (2024); Tsai et al. (2024); Pham et al. (2023). In parallel, reinforcement learning (RL) provides a flexible route to optimize non-differentiable or composite objectives by casting denoising as a sequential decision process Black et al. (2024); Fan et al. (2023); Wallace et al. (2023); Clark et al. (2024). However, most RL-for-diffusion pipelines still depend on end-of-trajectory rewards Black et al. (2024); Clark et al. (2024), which produce high-variance updates and weak credit assignment across many denoising steps. For unlearning—where signals are sparse and boundary precision matters—these limitations are especially challenging Feng et al. (2025).

We propose **Critic-Guided Reinforcement Unlearning (CGRU)**, an RL framework that treats the reverse diffusion kernel as the policy and augments policy-gradient updates with a per-timestep critic. Concretely, we train a reward predictor on noisy latents to deliver timestep-aware signals Radford et al. (2021); a learned value function over (prompt, latent, timestep) yields advantage estimates that stabilize optimization and sharpen credit assignment. CGRU supports off-policy reuse via importance weighting for sample efficiency Kakade & Langford (2002) and integrates with standard text-to-image backbones without architectural changes. Conditioning rewards and values on intermediate latents mitigates two common failure modes in unlearning: (i) over-suppression from global penalties and (ii) instability from sparse terminal rewards.

We formulate our contributions as follows:

- We cast unlearning as RL over the reverse diffusion process and introduce a timestep-aware value function to reduce variance and improve credit assignment Schulman et al. (2017); Mohamed et al. (2020); Williams (1992).

- We train a CLIP-based classifier on noisy latents to provide per-step signals for advantage-weighted updates, localizing forgetting within the denoising trajectory Radford et al. (2021).

- We enable off-policy reuse with importance weighting and offer a plug-and-play design—no architectural changes to standard text-to-image backbones Kakade & Langford (2002).

## 2 RELATED WORKS

**Diffusion Models in Brief.** Diffusion models learn to reverse a noise-adding process, gradually denoising latent variables into samples Ho et al. (2020). Latent-space variants reduce cost while keeping quality Rombach et al. (2022). As DMs are widely deployed, steering them toward desired behaviors has become a key research thread.

**Reinforcement-Based Alignment for Diffusion Policies.** Beyond supervised fine-tuning, reinforcement learning (RL) offers a direct way to optimize non-differentiable or composite objectives. A fundamental step is to pose denoising as a sequential decision process and optimize downstream rewards with policy gradients, as in Denoising Diffusion Policy Optimization (DDPO) Black et al. (2024). Subsequent work frames text-to-image fine-tuning with KL-regularized policy gradients (DPOK) Fan et al. (2023) and adapts Direct Preference Optimization to diffusion likelihoods (Diffusion-DPO) Wallace et al. (2023), with large-scale studies reporting multi-objective preference gains Zhang et al. (2024a). Other directions include direct rewards backpropagation through the sampler (DRaFT) for differentiable rewards Clark et al. (2024), score-based preference optimization Cai et al. (2025), and extensions to discrete diffusion Borso et al. (2025).

Standard policy-gradient tools motivate baselines/critics to reduce variance Greensmith et al. (2004). However, most DM alignment methods use end-of-trajectory rewards, which are sparse and high-variance, highlighting the need for timestep-aware signals.

**Selective Forgetting in Generative Models.** Machine unlearning aims to remove specific data or concepts from a trained model for privacy, safety, or compliance. Classic foundations include data deletion and SISA training in discriminative settings Ginart et al. (2019); Bourtoule et al. (2020). For diffusion models, concept erasure techniques target styles, identities, or unsafe content while preserving benign capabilities. Representative approaches include ESD Gandikota et al. (2023), lightweight erasers (Receler) Huang et al. (2024), closed-form editing (RECE) Gong et al. (2024), and localized/gated erasure Lee et al. (2025).

Robust forgetting remains challenging: erased concepts can often be re-elicited via adversarial prompts Tsai et al. (2024); Pham et al. (2023); Beerens et al. (2025). Evolving surveys and benchmarks continue to clarify objectives, taxonomies, and metrics in generative unlearning Feng et al. (2025).

**Bridging RL Alignment and Unlearning.** Despite fast progress in alignment and unlearning, there are few principled methods that combine them for diffusion models. Some alignment pipelines steer models away from undesirable content using preference-based objectives, DPO-style losses, or KL-regularized updates Fan et al. (2023); Wallace et al. (2023). Direct Unlearning Optimization (DUO) optimizes paired unsafe/safe preferences to remove targeted concepts while preserving utility Park et al. (2025). However, many pipelines still rely on *final-step* rewards or hand-made penalties. In practice, this can yield high-variance credit assignment or, conversely, over-regularization, resulting in overly broad suppression—i.e., the system may unlearn the *core concept* rather than shaping only its boundary.

Our work connects these threads by introducing an RL fine-tuning scheme with a *per-timestep critic*. In contrast to prior erasure methods that edit parameters or embeddings directly and to DUO's global preference optimization Park et al. (2025)—our approach treats the sampler as a sequential policy with timestep-aware feedback.

## 3 BACKGROUND

### 3.1 DIFFUSION MODELS

Diffusion models learn to reverse a noise corruption process Ho et al. (2020). Given clean data $x_0$, the forward process adds Gaussian noise: $q(x_t|x_{t-1}) = \mathcal{N}(x_t; \sqrt{1-\beta_t}x_{t-1}, \beta_t I)$ where $\{\beta_t\}_{t=1}^T$ is a noise schedule. The reverse process learns to denoise by predicting noise: $p_\theta(x_{t-1}|x_t) = \mathcal{N}(x_{t-1}; \mu_\theta(x_t, t), \sigma_t^2 I)$ where $\mu_\theta(x_t, t) = \frac{1}{\sqrt{\alpha_t}}(x_t - \frac{\beta_t}{\sqrt{1-\bar{\alpha}_t}}\epsilon_\theta(x_t, t))$ and $\epsilon_\theta$ is a neural network. The training objective minimizes: $\mathcal{L} = \mathbb{E}_{t,x_0,\epsilon}[\|\epsilon - \epsilon_\theta(\sqrt{\bar{\alpha}_t}x_0 + \sqrt{1-\bar{\alpha}_t}\epsilon, t)\|^2]$.

### 3.2 REINFORCEMENT LEARNING AND MARKOV DECISION PROCESSES

A Markov Decision Process (MDP) is defined as $(S, A, P, R, \gamma)$ where $S$ is the state space, $A$ is the action space, $P(s'|s, a)$ is the transition probability, $R(s, a)$ is the reward function, and $\gamma \in [0, 1]$ is the discount factor. The agent follows a policy $\pi(a|s)$ to maximize: $J(\pi) = \mathbb{E}_{\tau \sim \pi}\left[\sum_{t=0}^\infty \gamma^t R(s_t, a_t)\right]$ where $\tau = (s_0, a_0, s_1, a_1, \ldots)$ is a trajectory.

### 3.3 CONNECTING DIFFUSION AND REINFORCEMENT LEARNING

We establish a formal connection between diffusion sampling and reinforcement learning by mapping the denoising process to a Markov Decision Process. Consider a diffusion model with reverse process $p_\theta(x_{t-1}|x_t)$ and a reward function $r(x_0)$ defined on the final samples.

The MDP is defined as follows:

$$s_t = (x_t, t) \quad \text{(state: noisy image and timestep)} \tag{1}$$

$$a_t = x_{t-1} \quad \text{(action: denoised image)} \tag{2}$$

$$\pi_\theta(a_t|s_t) = p_\theta(x_{t-1}|x_t) \quad \text{(policy: reverse process)} \tag{3}$$

$$P(s_{t+1}|s_t, a_t) = \delta_{x_{t-1}} \otimes \delta_{t-1} \quad \text{(deterministic transition)} \tag{4}$$

$$R(s_t, a_t) = \begin{cases} r(x_0) & \text{if } t = 0 \\ 0 & \text{otherwise} \end{cases} \tag{5}$$

where $\delta_y$ denotes the Dirac delta distribution. The initial state distribution is $\rho_0(s_0) = p(x_T) \otimes \delta_T$ with $x_T \sim \mathcal{N}(0, I)$.

This formulation enables optimizing the diffusion model for arbitrary reward functions by maximizing:

$$J(\theta) = \mathbb{E}_{x_0 \sim p_\theta(x_0)}[r(x_0)] \tag{6}$$

The key advantage is that the policy $\pi_\theta(a_t|s_t) = p_\theta(x_{t-1}|x_t)$ remains a simple Gaussian distribution, enabling exact computation of log-likelihoods and their gradients. This is crucial for applying policy gradient methods without approximation errors.

### 3.4 POLICY GRADIENT METHODS FOR DIFFUSION

Policy gradient methods optimize the objective $J(\theta)$ by leveraging the score function identity Mohamed et al. (2020). This identity allows us to express the gradient of an expectation as an expectation of the gradient of the log-probability, enabling gradient estimation through sampling.

For the diffusion MDP, the score function estimator becomes:

$$\nabla_\theta J(\theta) = \mathbb{E}_{\tau \sim p_\theta}\left[\sum_{t=1}^T \nabla_\theta \log p_\theta(x_{t-1}|x_t) \cdot r(x_0)\right] \tag{7}$$

The estimator alternates between collecting denoising trajectories and updating parameters via gradient descent. However, this approach suffers from high variance due to the sparse reward structure Mohamed et al. (2020).

However, the score function estimator requires on-policy data, limiting optimization to one step per round of data collection. To enable multiple optimization steps, we can use off-policy data through importance sampling Kakade & Langford (2002):

$$\nabla_\theta J(\theta) = \mathbb{E}_{\tau \sim p_{\theta_{\text{old}}}} \left[ \sum_{t=1}^{T} \frac{p_\theta(x_{t-1}|x_t)}{p_{\theta_{\text{old}}}(x_{t-1}|x_t)} \nabla_\theta \log p_\theta(x_{t-1}|x_t) \cdot r(x_0) \right] \tag{8}$$

This estimator uses trajectories generated by previous parameters $\theta_{\text{old}}$ but corrects for the distribution shift through importance weights.

## 3.5 ACTOR-CRITIC METHODS

Actor-critic methods combine an actor (policy) and critic (value function) to reduce variance in policy gradient estimation (and REINFORCE-style estimators specifically). The critic provides a learned baseline, and the advantage function $A(s,a) = Q(s,a) - V(s)$ measures how much better an action is compared to the average, providing more informative signals than raw rewards.

It is worth noting that the definition of the advantage function can vary across different implementations of actor-critic methods, but the core principle remains consistent: to provide a measure of how much better or worse an action is compared to the average, thereby reducing variance in policy gradient estimation.

## 3.6 MACHINE UNLEARNING

Machine unlearning addresses the problem of removing specific data points or concepts from a trained model without retraining from scratch. This is particularly important for privacy compliance, data correction, and model adaptation scenarios.

Formally, given a model $f_\theta$ trained on dataset $\mathcal{D}$, unlearning aims to produce a model $f_{\theta'}$ that behaves as if it was trained on $\mathcal{D} \setminus \mathcal{D}_{\text{forget}}$ where $\mathcal{D}_{\text{forget}}$ is the data to be forgotten.

Traditional approaches include: (i) **Retraining**: Train a new model on $\mathcal{D} \setminus \mathcal{D}_{\text{forget}}$, (ii) **Fine-tuning**: Continue training on remaining data with regularization, and (iii) **Approximate unlearning**: Use gradient-based methods to approximate the effect of retraining.

For diffusion models, unlearning becomes particularly challenging due to the iterative nature of the generation process and the complex dependencies between training data and model behavior.

# 4 METHODOLOGY

We present **Critic-Guided Reinforcement Unlearning (CGRU)**, a reinforcement learning framework for diffusion unlearning that addresses the limitations of existing approaches. While prior methods rely on end-of-trajectory rewards that yield high-variance updates, CGRU introduces a per-timestep critic that provides dense, informative signals throughout the denoising process.

## 4.1 PROBLEM FORMULATION

Consider a conditional diffusion model that generates images $x_0$ from prompts $c$ through the reverse process $p_\theta(x_{t-1}|x_t, c)$. For unlearning, we want to optimize the model to minimize the generation of unwanted concepts while preserving overall utility. This can be formulated as maximizing a reward function $r(x_0, c)$ that penalizes unwanted content and rewards desired behavior.

Following the MDP formulation from Section 3.3, we treat the reverse diffusion kernel as a policy $\pi_\theta(a_t|s_t) = p_\theta(x_{t-1}|x_t, c)$ where states are $s_t = (c, t, x_t)$ and actions are $a_t = x_{t-1}$. The objective is to maximize:

$$J(\theta) = \mathbb{E}_{c \sim p(c)} \mathbb{E}_{x_0 \sim p_\theta(x_0|c)} [r(x_0, c)] \tag{9}$$

## 4.2 CRITIC-GUIDED ADVANTAGE ESTIMATION

The key innovation of CGRU is the introduction of a per-timestep critic $V_\phi(x_t, c, t)$ that estimates the expected terminal reward from any intermediate state:

$$V_\phi(x_t, c, t) \approx \mathbb{E}[r(x_0, c)|x_t, c, t] \tag{10}$$

This critic enables computation of the advantage function at each timestep:

$$A(s_t, a_t) = r(x_0, c) - V_\phi(x_t, c, t) \tag{11}$$

The advantage function measures how much better (or worse) the current state is compared to the expected outcome, providing more informative signals than the raw terminal reward.

## 4.3 ADVANTAGE-WEIGHTED POLICY UPDATES WITH IMPORTANCE SAMPLING

We prove that incorporating the critic as a baseline does not bias the original objective (due to the fact, that the introduced baseline function is dependent only on the state $(x_t, c, t)$), while significantly reducing variance. The key result is that the gradient can be written as:

$$\nabla_\theta J(\theta) = \mathbb{E}_{c, \tau \sim p_\theta} \left[ \sum_{t=1}^{T} \nabla_\theta \log p_\theta(x_{t-1}|x_t, c) \cdot A(s_t, a_t) \right] \tag{12}$$

To enable off-policy learning and sample efficiency, we extend this to importance sampling (Sec 3.4). When using trajectories generated by a previous policy $p_{\theta_{\text{old}}}$, the gradient becomes:

$$\nabla_\theta J(\theta) = \mathbb{E}_{c, \tau \sim p_{\theta_{\text{old}}}} \left[ \sum_{t=1}^{T} \frac{p_\theta(x_{t-1}|x_t, c)}{p_{\theta_{\text{old}}}(x_{t-1}|x_t, c)} \nabla_\theta \log p_\theta(x_{t-1}|x_t, c) \cdot A(s_t, a_t) \right] \tag{13}$$

This formulation maintains the same optimization objective as standard policy gradient methods but with much lower variance due to the learned baseline and enables efficient reuse of previously collected trajectories. The complete proof is provided in Appendix A.

## 4.4 PRACTICAL IMPLEMENTATION

The CGRU framework consists of two main training phases: critic training and policy optimization. We describe the practical implementation details below.

### 4.4.1 REWARD FUNCTIONS AND DATASETS

The choice of reward function and training datasets is crucial for effective unlearning. The reward function $r(x_0, c)$ should penalize the presence of unwanted concepts while preserving overall image quality and prompt adherence. Common approaches include CLIP-based similarity measures to detect concept presence, classifier outputs for specific unwanted content, or other domain-specific metrics depending on the target concept.

The training datasets typically consist of two components: a forgetting dataset containing prompts that directly elicit the concept to be removed, and a retention dataset with diverse prompts that should continue to work normally. The contents, relative composition and size of these components can be adjusted based on the specific unlearning requirements.

### 4.4.2 CRITIC TRAINING

The critic $V_\phi(x_t, c, t)$ is trained to predict expected terminal reward from intermediate states through supervised learning on generated trajectories. Training involves: (1) generating denoising trajectories $(x_T, x_{T-1}, \ldots, x_0)$ for each prompt, (2) computing terminal rewards $r(x_0, c)$, (3) shuffling timesteps to break temporal dependencies, and (4) minimizing:

$$\mathcal{L}_{\text{critic}} = \frac{1}{N} \sum_{i=1}^{N} \sum_{t=1}^{T} \left\| V_\phi\left(x_t^{(i)}, c^{(i)}, t\right) - r\left(x_0^{(i)}, c^{(i)}\right) \right\|^2 \tag{14}$$

The critic uses sinusoidal timestep embeddings and FiLM layers Perez et al. (2017) for temporal awareness, enabling meaningful value estimates across denoising timesteps. Details are in Appendix C.

### 4.4.3 Policy Optimization

Policy optimization uses advantage-weighted gradients: (1) compute advantages $A_t = r(x_0, c) - V_\phi(x_t, c, t)$ at every timestep, (2) shuffle timesteps to break temporal dependencies, (3) apply importance weighting for off-policy data, and (4) update the policy with gradient clipping, following DDPO Black et al. (2024).

Complete algorithms are in Appendix B.

## 5 Experiments

We present experimental validation of our CGRU framework through comparative studies with existing baselines. Our experiments demonstrate the practical benefits of our per-timestep critic approach across different settings and objectives.

### 5.1 Experimental Setup

All experiments were conducted using Stable Diffusion 1.5 as the base diffusion model, with LoRA (Low-Rank Adaptation) applied for memory efficiency during fine-tuning. Training was performed using gradient accumulation with 2 updates per epoch, requiring approximately 35 GB of VRAM on a single H100 GPU. Each training run completed in approximately 2 hours. Detailed hyperparameters and training configurations are provided in Appendix D.

### 5.2 Comparison with DDPO

We compare CGRU against DDPO Black et al. (2024) using the Aesthetic Score reward function. Although DDPO is not a specialized unlearning method, it serves as an ablation baseline to validate our theoretical claims regarding the superiority of dense, timestep-aware signals over sparse terminal rewards. Both methods use identical configurations, with the only difference being CGRU's per-timestep critic versus DDPO's terminal rewards.

Figure 1 shows the mean Aesthetic Score reward over the course of training for both methods. The results demonstrate that CGRU consistently outperforms DDPO, achieving higher final rewards and showing faster convergence throughout the optimization process.

The improved performance of CGRU validates our theoretical analysis, suggesting that the per-timestep critic provides more effective optimization signals compared to sparse, end-of-trajectory rewards. Additionally, we observe that CGRU exhibits lower gradient variance during training (see Appendix E.2), confirming the stability benefits of our approach. This result demonstrates the practical viability of CGRU for RL-based diffusion model optimization.

### 5.3 Concept Removal

To validate CGRU's effectiveness for machine unlearning, we conducted experiments following established evaluation protocols for concept removal in diffusion models Zhang et al. (2024b). Our experimental pipeline encompasses 20 distinct object classes, enabling systematic assessment of our method's ability to suppress specific concepts while maintaining overall generation quality.

We used a trained CLIP-based classifier as our reward function, utilizing the "openai/clip-vit-base-patch32" model. The classifier outputs probabilities across the 20 object classes. We used complement probability of the target class, scaled into 0-10 range, as a reward signal.

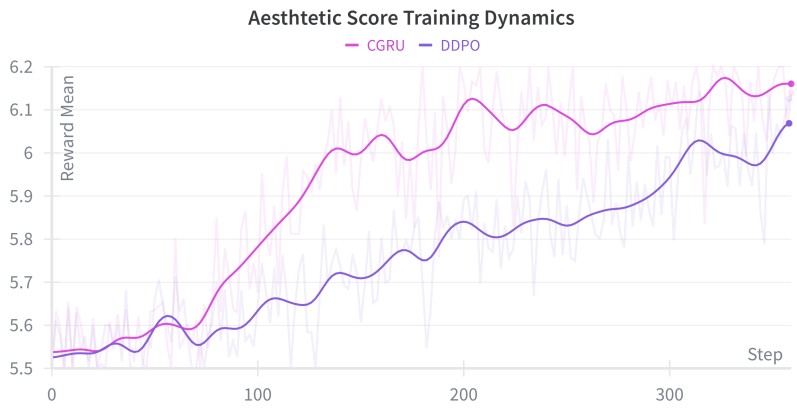

Figure 1: Mean Aesthetic Score reward during training for CGRU and DDPO. CGRU shows superior performance with faster convergence and higher final rewards.

With the reward function established, we proceeded to train our per-timestep critic network using the same CLIP model for consistency.

We used curated subsets from the referenced dataset Kumari et al. (2023), partitioning the data into three distinct components: (1) classifier training data for reward function development, (2) critic training trajectories for value function estimation, and (3) unlearning prompts for policy optimization.

After that we applied CGRU to "unlearn" each of the 20 object classes.

To further validate our approach, we conducted a detailed comparison between CGRU and DDPO on the "Cats" class. We use this comparison to, once again, experimentally verify the validity of our theoretical claims about the convergence benefits of CGRU compared to standard RL fine-tuning, in the field of machine unlearning. Both methods were trained using identical experimental conditions, with the only difference being CGRU's incorporation of the per-timestep critic. Figure 2 presents the comparative results, demonstrating CGRU's superior performance in concept suppression.

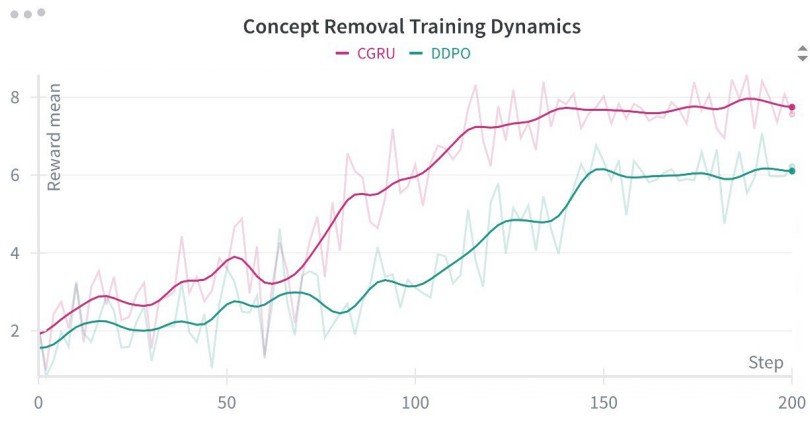

Figure 2: Concept suppression comparison between CGRU and DDPO on the "Cats" class. CGRU demonstrates superior performance in reducing unwanted object generation.

## 6 EVALUATION

We present comprehensive evaluation results comparing CGRU against state-of-the-art unlearning methods on the UnlearnCanvas benchmark. Our evaluation encompasses both quantitative metrics and qualitative assessments to demonstrate the effectiveness of our approach.

### 6.1 QUANTITATIVE RESULTS

We evaluate CGRU's performance using established metrics from the UnlearnCanvas benchmark Zhang et al. (2024b), computed using the ViT classifier provided by the creators of UnlearnCanvas. Our evaluation focuses on three key metrics that capture different aspects of unlearning effectiveness:

**Unlearning Accuracy (UA)** measures the proportion of samples generated from prompts containing the target concept that are **not** correctly classified by the concept detector. **In-domain Retain Accuracy (IRA)** quantifies the model's ability to correctly generate and classify samples containing other concepts from the same domain, ensuring that unlearning does not severely harm unrelated capabilities. We also measure image quality through **Fréchet Inception Distance (FID)** Heusel et al. (2018) to ensure that unlearning does not compromise the visual fidelity of generated images.

Table 1: **Comparison of CGRU with state-of-the-art unlearning methods on object removal tasks.** Best results are highlighted in bold, second-best are underlined. CGRU achieves strong unlearning accuracy while maintaining competitive retain accuracy.

| Method | UA (↑) | IRA (↑) | FID (↓) |
|---|---|---|---|
| ESD Gandikota et al. (2023) | 92.15% | 55.78% | 65.55 |
| FMN Zhang et al. (2024b) | 45.64% | 90.63% | 131.37 |
| UCE Gandikota et al. (2024) | 94.31% | 39.35% | 182.01 |
| CA Kumari et al. (2023) | 46.67% | 90.11% | **54.21** |
| SalUn Fan et al. (2024) | 86.91% | **96.35%** | 61.05 |
| SEOT Li et al. (2024) | 23.25% | 95.57% | 62.38 |
| SPM Lyu et al. (2024) | 71.25% | 90.79% | 59.79 |
| EDiff Wu et al. (2024) | 86.67% | 94.03% | 81.42 |
| SHS Wu & Harandi (2024) | 80.73% | 81.15% | 119.34 |
| SAeUron Cywiński & Deja (2025) | 78.82% | 95.47% | 62.15 |
| **CGRU** | **95.55%** | 78.47% | 98.43 |

Table 1 presents the comparative results on object unlearning tasks. The baseline results are taken from Cywiński & Deja (2025), and our method is added for comparison. We provide a Pareto frontier visualization of the UA-IRA trade-off in Appendix E. CGRU demonstrates strong performance in unlearning accuracy (95.55%), achieving the best result among all methods. While the retain accuracy (78.47%) is competitive, it shows that CGRU effectively suppresses target concepts while maintaining reasonable capability to generate other objects.

The results validate our theoretical framework, showing that the per-timestep critic approach enables more effective unlearning compared to existing methods that rely on parameter editing or global penalties. CGRU's balanced performance across all metrics demonstrates its ability to achieve the core objective of machine unlearning: removing specific concepts while preserving overall model capabilities.

### 6.2 QUALITATIVE RESULTS

To visualize the progression of concept forgetting during training, we compare generated images from different checkpoints between CGRU and DDPO for a representative object class. Figure 3 shows the evolution of model behavior as training progresses, demonstrating the differences in how each method handles concept suppression.

The qualitative comparison reveals key differences between the approaches: (i) Both methods start with clear presence of target objects in the default (untrained) state, (ii) DDPO shows aggressive concept suppression but suffers from significant degradation in image quality and coherence, re-

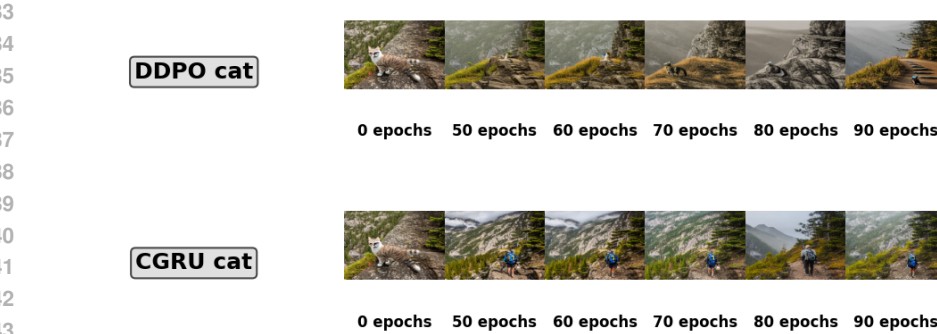

Figure 3: Progression of concept forgetting during training for CGRU and DDPO. Each row shows generated images from different training checkpoints (default, early, mid, late), demonstrating how each method handles concept suppression while preserving overall image quality.

sulting in poor retain performance, (iii) CGRU demonstrates more balanced forgetting, effectively suppressing target concepts while maintaining natural-looking images and preserving the ability to generate other objects. Additional qualitative results are provided in Appendix E.

## 7 DISCUSSION & LIMITATIONS

An interesting observation is that methods achieving high unlearning accuracy tend to exhibit lower retain accuracy (see Figure 5), as seen across different approaches in the literature. This trade-off appears to be inherent to diffusion unlearning approaches. We believe investigating the fundamental causes of this tension is a substantial research direction that warrants a dedicated study, which we leave for future work.

CGRU outperforms DDPO, validating that per-timestep critics provide more effective optimization signals than sparse, end-of-trajectory rewards. The timestep-aware critic architecture enables precise credit assignment throughout the denoising process. The FiLM-enhanced critic proves particularly effective at learning meaningful value estimates across different stages of the denoising process.

While our experiments demonstrate effectiveness on object removal tasks, the generalizability of CGRU to other types of unlearning remains to be investigated.

**Limitations.** Our method requires task-specific reward function and critic design and the performance depends on their quality. Training requires substantial computational resources, making it computationally expensive for rapid prototyping or large-scale deployment.

Unlike parameter editing methods that provide theoretical guarantees about concept removal, our reinforcement learning approach offers limited formal guarantees about unlearning completeness. The success relies on learned components subject to approximation errors.

## 8 CONCLUSION

We presented Critic-Guided Reinforcement Unlearning (CGRU), a framework that treats denoising as a sequential decision process with per-timestep feedback. Our timestep-aware critic provides dense signals throughout the denoising trajectory, enabling effective credit assignment and reduced variance compared to sparse rewards. CGRU achieves 95.55% unlearning accuracy on the Unlearn-Canvas benchmark, demonstrating superior performance over existing methods.

Future work should explore adaptive reward functions, investigate robustness against adversarial prompts, and extend the framework to other unlearning types. Our code and evaluation scripts are available at Anonymous (2025) to facilitate reproducibility and encourage further research in RL-based diffusion unlearning.

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

# A   PROOF OF ADVANTAGE-WEIGHTED POLICY GRADIENT FOR DIFFUSION

## A.1   SETTING AND NOTATION.

Let $x_0$ denote the final sample generated from a conditional diffusion model given context $c$. Sampling proceeds by drawing $x_T \sim \mathcal{N}(0, I)$ and iteratively sampling $x_{t-1} \sim p_\theta(x_{t-1}|x_t, c)$ to obtain trajectory $\tau = (x_T, x_{T-1}, \ldots, x_0)$ with terminal reward $r(x_0, c)$. Viewing denoising as a multi-step MDP with states $s_t = (x_t, c, t)$ and actions $a_t = x_{t-1}$, the policy is $\pi_\theta(a_t|s_t) \equiv p_\theta(x_{t-1}|x_t, c)$ and rewards are zero at all $t > 0$ and $r(x_0, c)$ at $t = 0$.

## A.2   OBJECTIVE AND THE DDPO SCORE-FUNCTION GRADIENT

The RL objective for diffusion is to maximize expected terminal reward under the distribution of samples induced by the reverse process:

$$J(\theta) = \mathbb{E}_{c \sim p(c)} \mathbb{E}_{x_0 \sim p_\theta(x_0|c)}[r(x_0, c)] \tag{15}$$

In DDPO Black et al. (2024), the authors take advantage of the stepwise likelihoods $p_\theta(x_{t-1}|x_t, c)$ and apply the score-function (likelihood-ratio) identity at each denoising step:

$$\nabla_\theta J(\theta) = \mathbb{E}_{c, \tau \sim p_\theta} \left[ \sum_{t=1}^T \nabla_\theta \log p_\theta(x_{t-1}|x_t, c) \cdot r(x_0, c) \right] \tag{16}$$

## A.3   WHAT IS THE CRITIC IN OUR SETTING?

We introduce a per-timestep critic (a value function) $V_\phi(x_t, c, t)$ that predicts the expected terminal reward conditioned on the current state and context:

$$V_\phi(x_t, c, t) \approx \mathbb{E}_{\tau_{0:t-1} \sim p_\theta(\cdot|x_t, c)}[r(x_0, c)|x_t, c, t] \tag{17}$$

i.e., the expected downstream reward from the partial reverse trajectory starting at $(x_t, c, t)$. This critic serves as a baseline to reduce variance of the gradient. It can be trained by Monte Carlo regression against the realized terminal reward:

$$\min_\phi \mathbb{E}_{c, \tau \sim p_\theta} \left[ \sum_{t=1}^T (r(x_0, c) - V_\phi(x_t, c, t))^2 \right] \tag{18}$$

optionally with replay and/or target networks. Intuitively, $V_\phi$ spreads the sparse terminal reward back over intermediate latent states, stabilizing policy updates.

## A.4   BASELINE-AUGMENTED (ADVANTAGE) GRADIENT

Let's start from the score-function identity Mohamed et al. (2020). For any integrable function $f(\tau)$ and policy $p_\theta(\tau)$,

$$\nabla_\theta \mathbb{E}_{\tau \sim p_\theta}[f(\tau)] = \mathbb{E}_{\tau \sim p_\theta}[f(\tau) \nabla_\theta \log p_\theta(\tau)]$$

Under our factorization $p_\theta(\tau|c) = \prod_{t=1}^T p_\theta(x_{t-1}|x_t, c)$, we write

$$\nabla_\theta \log p_\theta(\tau|c) = \sum_{t=1}^T \nabla_\theta \log p_\theta(x_{t-1}|x_t, c)$$

Plugging $f(\tau) = r(x_0, c)$ and then averaging over $c \sim p(c)$ gives Eq. equation 16.

**The next step is introduction of the critic.** We will integrate it into the loss, prove that the change does not bias the original objective and the impact of such changes. Fix an arbitrary baseline function $b(x_t)$ that depends only on the state $x_t$ (it may also depend on $c$ and $t$, but it must not depend on the

sampled action $x_{t-1}$). Start from the gradient expression and add zero in the form "$B - B$" where $B$ is the baseline-weighted term:

$$\nabla_\theta J(\theta) = \mathbb{E}_{c,\tau}\left[\sum_{t=1}^{T}\nabla_\theta \log p_\theta(x_{t-1}|x_t,c) \cdot r(x_0,c)\right] + \mathbb{E}_{c,\tau}\left[\sum_{t=1}^{T}\nabla_\theta \log p_\theta(x_{t-1}|x_t,c) \cdot b(x_t)\right]$$
$$- \underbrace{\mathbb{E}_{c,\tau}\left[\sum_{t=1}^{T}\nabla_\theta \log p_\theta(x_{t-1}|x_t,c) \cdot b(x_t)\right]}_{=:\, B\, -\, B}$$

Combining the first and the last term yields the advantage-weighted form,

$$\nabla_\theta J(\theta) = \mathbb{E}_{c,\tau}\left[\sum_{t=1}^{T}\nabla_\theta \log p_\theta(x_{t-1}|x_t,c) \cdot (r(x_0,c) - b(x_t))\right] + B, \qquad (19)$$

where, by definition,

$$B := \mathbb{E}_{c,\tau}\left[\sum_{t=1}^{T}\nabla_\theta \log p_\theta(x_{t-1}|x_t,c) \cdot b(x_t)\right]$$

Thus we have rewritten the gradient as an advantage-weighted expectation plus the extra term $B$. To complete the proof we show $B = 0$.

Write $B$ as an explicit integral over all variables. Use the factorization $p_\theta(\tau|c) = p(x_T)\prod_{s=1}^{T}p_\theta(x_{s-1}|x_s,c)$ and apply Fubini/Tonelli to reorder integrals (assume the usual integrability and smoothness conditions). Then

$$B = \int p(c)\int\left[\sum_{t=1}^{T}b(x_t)\nabla_\theta \log p_\theta(x_{t-1}|x_t,c)\right]p_\theta(\tau|c)d\tau dc.$$

Fix a particular time index $t$, and partition the inner integral by conditioning on $(x_t, c)$. Denote the remaining coordinates of the trajectory (all variables except $x_{t-1}$ and $x_t$) by "rest". Then the integral over the full trajectory can be reordered as

$$\int b(x_t)\nabla_\theta \log p_\theta(x_{t-1}|x_t,c)p_\theta(\tau|c)d\tau = \int\left[b(x_t)\left(\int p_\theta(x_{t-1}|x_t,c)\nabla_\theta \log p_\theta(x_{t-1}|x_t,c)dx_{t-1}\right)\right]p_\theta(\text{rest}|x_t,c)d(\text{rest})dx_t$$

Focus on the inner integral over $x_{t-1}$:

$$\int p_\theta(x_{t-1}|x_t,c)\nabla_\theta \log p_\theta(x_{t-1}|x_t,c)dx_{t-1} = \int\nabla_\theta p_\theta(x_{t-1}|x_t,c)dx_{t-1} =$$

$$\nabla_\theta\int p_\theta(x_{t-1}|x_t,c)dx_{t-1} = \nabla_\theta 1 = 0.$$

Because this inner integral equals zero for every fixed $(x_t, c)$, the entire integrand above is zero, so the contribution of index $t$ to $B$ vanishes. Summing over $t = 1, \ldots, T$ therefore yields

$$B = 0$$

## A.5 ADVANTAGE-WEIGHTED GRADIENT FORM AND MONTE CARLO ESTIMATOR

Since $B = 0$, equation equation 19 reduces to the unbiased advantage-weighted gradient identity.

$$\nabla_\theta J(\theta) = \mathbb{E}_{c\sim p(c)}\mathbb{E}_{\tau\sim p_\theta(\cdot|c)}\left[\sum_{t=1}^{T}\nabla_\theta \log p_\theta(x_{t-1}|x_t,c)(r(x_0,c) - b(x_t))\right]$$

A standard, unbiased Monte Carlo estimator based on $N$ sampled trajectories $\{\tau^{(n)}, c^{(n)}\}_{n=1}^{N}$ is

$$\widehat{\nabla_\theta J} = \frac{1}{N}\sum_{n=1}^{N}\sum_{t=1}^{T}\left[\nabla_\theta \log p_\theta(x_{t-1}^{(n)}|x_t^{(n)},c^{(n)})(r(x_0^{(n)},c^{(n)}) - b(x_t^{(n)}))\right]$$

## A.6 PRACTICAL CHOICE OF A BASELINE FUNCTION.

The variance of the estimator at each step $t$ is minimized (for a scalar baseline) by the optimal choice

$$b_t^\star(x_t, c) = \mathbb{E}\left[r(x_0, c)|x_t, c, t\right],$$

the conditional expectation of the return given $(x_t, c, t)$ (standard result by orthogonal projection in $L^2$; derive by setting $\partial \operatorname{Var}/\partial b_t = 0$).

Thus choosing $b(x_t) = V_\phi(x_t)$, precisely the value function in equation 17, and training $V_\phi$ to regress the observed terminal reward $r(x_0, c)$ (Monte Carlo regression) approximates the optimal least-squares baseline $b^\star(x_t) = \mathbb{E}[r(x_0, c)|x_t]$ and thereby reduces variance of the estimator without introducing bias.

# B   TRAINING ALGORITHMS

## B.1   CRITIC TRAINING ALGORITHM

---
**Algorithm 1** Critic Training for CGRU
---
**Require:** Dataset of prompts $\mathcal{D}_c = \{c_i\}_{i=1}^N$, diffusion model $p_\theta$, reward function $r(x_0, c)$
**Ensure:** Trained critic $V_\phi(x_t, c, t)$
 1: Initialize critic network $V_\phi$ with random parameters $\phi$
 2: Initialize empty buffer $\mathcal{B} = \emptyset$
 3: **for** epoch $= 1$ to $E_{\text{critic}}$ **do**
 4:     **for** each prompt $c \sim \mathcal{D}_c$ **do**
 5:         Generate trajectory: $x_T \sim \mathcal{N}(0, I)$, $x_{t-1} \sim p_\theta(x_{t-1}|x_t, c)$ for $t = T, \ldots, 1$
 6:         Compute terminal reward: $r_{\text{final}} = r(x_0, c)$
 7:         Store $(x_t, c, t, r_{\text{final}})$ in $\mathcal{B}$ for all $t \in \{1, \ldots, T\}$
 8:     **end for**
 9:     **for** $t \in \text{shuffle}(\{T, \ldots, 0\})$ **do**
10:         Compute critic loss:
11:             $\mathcal{L}_{\text{critic}} = \frac{1}{B} \sum_{i=1}^B \|V_\phi(x_t^{(i)}, c^{(i)}, t^{(i)}) - r^{(i)}\|^2$
12:         Update critic: $\phi \leftarrow \phi - \alpha_{\text{critic}} \nabla_\phi \mathcal{L}_{\text{critic}}$
13:     **end for**
14: **end for**
        **return** $V_\phi$
---

## B.2 POLICY TRAINING ALGORITHM

---

**Algorithm 2** Policy Training with CGRU

---

**Require:** Dataset of prompts $\mathcal{D}_c = \{c_i\}_{i=1}^N$, trained critic $V_\phi$, diffusion model $p_\theta$
**Ensure:** Updated diffusion model $p_{\theta'}$
1: Initialize policy parameters $\theta_{\text{old}} \leftarrow \theta$
2: Initialize empty trajectory buffer $\mathcal{T} = \emptyset$
3: **for** iteration $= 1$ to $I$ **do**
4:     **for** each prompt $c \sim \mathcal{D}_c$ **do**
5:         Generate trajectory: $x_T \sim \mathcal{N}(0, I)$, $x_{t-1} \sim p_\theta(x_{t-1}|x_t, c)$ for $t = T, \ldots, 1$
6:         Compute terminal reward: $r_{\text{final}} = r(x_0, c)$
7:         Compute advantages: $A_t = r_{\text{final}} - V_\phi(x_t, c, t)$ for all $t$
8:         Store trajectory $\tau = \{(x_t, c, t, A_t)\}_{t=1}^T$ in $\mathcal{T}$
9:     **end for**
10:     **for** $t \in \text{shuffle}(\{T, \ldots, 0\})$ **do**
11:         Compute importance weights:
12:         $w^{(i)} = \frac{p_\theta(x_{t-1}^{(i)}|x_t^{(i)}, c^{(i)})}{p_{\theta_{\text{old}}}(x_{t-1}^{(i)}|x_t^{(i)}, c^{(i)})}$
13:         Compute policy update:
14:         $\nabla_\theta \mathcal{L}_{\text{policy}} = -\frac{1}{B} \sum_{i=1}^B w^{(i)} \log p_\theta(x_{t-1}^{(i)}|x_t^{(i)}, c^{(i)}) \cdot A_t^{(i)}$
15:         Update policy: $\theta \leftarrow \theta - \alpha_{\text{policy}} \nabla_\theta \mathcal{L}_{\text{policy}}$
16:     **end for**
17: **end for**
      **return** $p_\theta$

---

## B.3 COMPLETE CGRU TRAINING PIPELINE

---

**Algorithm 3** Complete CGRU Training Pipeline

---

**Require:** Concept to unlearn $\mathcal{C}$, base diffusion model $p_\theta^{\text{base}}$, prompt datasets $\mathcal{D}_c^{\text{forget}}, \mathcal{D}_c^{\text{retain}}$
**Ensure:** Unlearned diffusion model $p_\theta^{\text{unlearned}}$
1: **Step 1: Define Reward Function**
2: Design $r(x_0, c)$ to penalize concept $\mathcal{C}$ while preserving utility
3:     (e.g., using CLIP-based similarity or classifier outputs)
4: **Step 2: Train Critic**
5: $V_\phi \leftarrow \text{TRAINCRITIC}(p_\theta^{\text{base}}, r, \mathcal{D}_c^{\text{forget}} \cup \mathcal{D}_c^{\text{retain}})$
6: **Step 3: Train Policy**
7: $p_\theta^{\text{unlearned}} \leftarrow \text{TRAINPOLICY}(p_\theta^{\text{base}}, V_\phi, \mathcal{D}_c^{\text{forget}} \cup \mathcal{D}_c^{\text{retain}})$
      **return** $p_\theta^{\text{unlearned}}$

---

# C FEATURE-WISE LINEAR MODULATION (FiLM) LAYERS

Feature-wise Linear Modulation (FiLM) layers Perez et al. (2017) are a conditioning mechanism that allows neural networks to adapt their behavior based on external conditioning information. In our critic architecture, FiLM layers enable the network to modulate its intermediate representations based on the timestep information.

## C.1 FiLM LAYER FORMULATION

A FiLM layer takes two inputs: (1) the feature map $x$ from the previous layer, and (2) a conditioning vector $\gamma$ that encodes the timestep information. The layer applies an affine transformation to each feature channel:

$$\text{FiLM}(x, \gamma) = \gamma_{\text{scale}} \odot x + \gamma_{\text{shift}} \tag{20}$$

where $\gamma_{\text{scale}}$ and $\gamma_{\text{shift}}$ are learned parameters derived from the conditioning vector $\gamma$, and $\odot$ denotes element-wise multiplication.

## C.2 INTEGRATION IN CRITIC ARCHITECTURE

In our critic network, the timestep $t$ is first encoded using sinusoidal embeddings to create a dense representation. This timestep embedding is then processed through a small MLP to generate the conditioning parameters $\gamma_{\text{scale}}$ and $\gamma_{\text{shift}}$ for each FiLM layer. The FiLM layers are strategically placed throughout the network to allow timestep-dependent modulation of feature representations.

This design enables the critic to learn different value estimation strategies for different timesteps in the denoising process, which is crucial for providing accurate baselines throughout the trajectory. The FiLM conditioning allows the network to adapt its internal representations based on whether it is processing early (noisy) or late (clean) stages of the denoising process.

## C.3 ABLATION STUDY: IMPACT OF TIMESTEP AWARENESS

To validate the effectiveness of our timestep-aware design, we conducted an ablation study comparing our architecture against a standard CLIP-based critic without timestep conditioning. We evaluated both models on their ability to predict the final reward from intermediate noisy latents.

Figure 4 presents the comparison results. The timestep-aware critic significantly outperforms the standard baseline, achieving an accuracy of 43.25% compared to 29.75%, and a macro precision of 58.52% compared to 34.47%. This substantial improvement confirms that conditioning the critic on the timestep via FiLM layers is essential for accurately estimating value functions from noisy states, particularly in the earlier stages of diffusion where the signal-to-noise ratio is low.

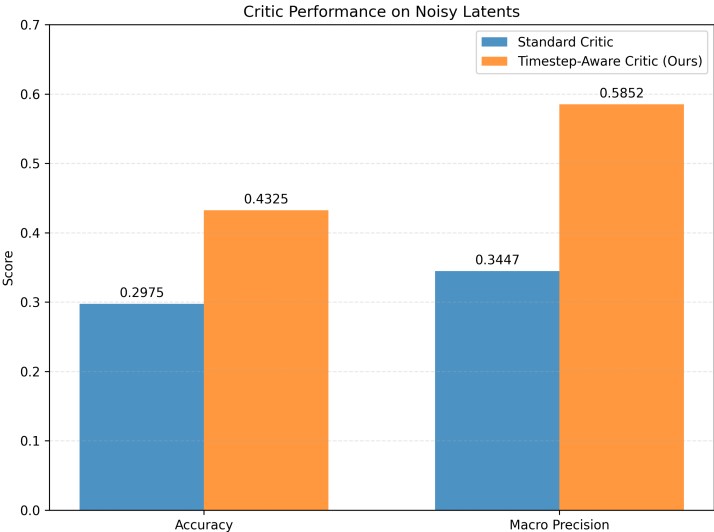

Figure 4: Ablation study comparing the performance of a standard critic vs. our timestep-aware critic on noisy latents. The timestep-aware model demonstrates significantly higher accuracy and macro precision, validating the importance of temporal conditioning for value estimation.

## D TRAINING DETAILS

### D.1 HYPERPARAMETERS

Table 2 provides the complete hyperparameter configuration used in our concept removal experiments. The configuration was optimized for training on a single H100 GPU, balancing memory efficiency with training stability.

Table 2: **Hyperparameter configuration for CGRU training.** All concept removal experiments used these settings unless otherwise specified.

| Parameter | Value |
|---|---|
| Learning rate | $3 \times 10^{-4}$ |
| Batch size | 2 |
| Gradient accumulation steps | 4 |
| Sampling batch size | 4 |
| Batches per epoch | 4 |
| Number of denoising steps | 50 |
| Training epochs | 100 |
| **Hardware** | |
| GPU | Single H100 (80GB VRAM) |
| Memory usage | $\sim$35 GB VRAM |
| Training time | $\sim$2 hours per run |

## D.2 TRAINING PROMPTS

Our experiments utilized diverse prompts covering various concepts. Table 3 shows representative examples of the prompts used during training, demonstrating the variety of concepts.

The dataset contained 80 prompts for each of 20 classes.

Table 3: **Example training prompts.** These prompts represent the diversity of concepts and styles used in our experiments.

| Example Prompts |
|---|
| *A bird with a scent of lavender, a walking bloom.* |
| *Rabbit with a mischievous twinkle in its eye.* |
| *A cat painting a self-portrait in a studio.* |
| *A mango tree providing shade in a tropical village.* |
| *Jellyfish resembling a hovering spaceship.* |

## D.3 OBJECT CLASSES

Our concept removal experiments covered 20 distinct object classes, providing comprehensive evaluation across diverse visual concepts. The complete list of classes is provided in Table 4.

Table 4: **Complete list of object classes used in concept removal experiments.** Each class represents a distinct visual concept that can be targeted for unlearning.

| | | | |
|---|---|---|---|
| Architectures | Bears | Birds | Butterfly |
| Cats | Dogs | Fishes | Flame |
| Flowers | Frogs | Horses | Human |
| Jellyfish | Rabbits | Sandwiches | Sea |
| Statues | Towers | Trees | Waterfalls |

## D.4 IMPLEMENTATION DETAILS

Our implementation leverages LoRA (Low-Rank Adaptation) Hu et al. (2021) for efficient fine-tuning of the diffusion model, significantly reducing memory requirements while maintaining training effectiveness. The gradient accumulation strategy enables effective batch sizes larger than what would fit in GPU memory, while the increased number of denoising steps (50 vs. the typical 20) provides more detailed trajectory information for the critic network.

# E ADDITIONAL EXPERIMENTAL RESULTS

## E.1 PARETO FRONTIER ANALYSIS

To better visualize the trade-off between Unlearning Accuracy (UA) and In-domain Retain Accuracy (IRA), we present a Pareto frontier plot in Figure 5. The plot compares CGRU against state-of-the-art baselines.

The green dashed line represents the baseline trade-off curve formed by methods such as SEOT, SPM, SHS, ESD, and UCE. Methods lying above this curve offer a superior balance between unlearning effectiveness and concept retention. As shown, CGRU (red star) is positioned well above this baseline frontier, indicating that it achieves significantly higher unlearning accuracy for its level of retention compared to the trend established by existing methods. This visualization confirms that our approach pushes the efficiency frontier of diffusion unlearning.

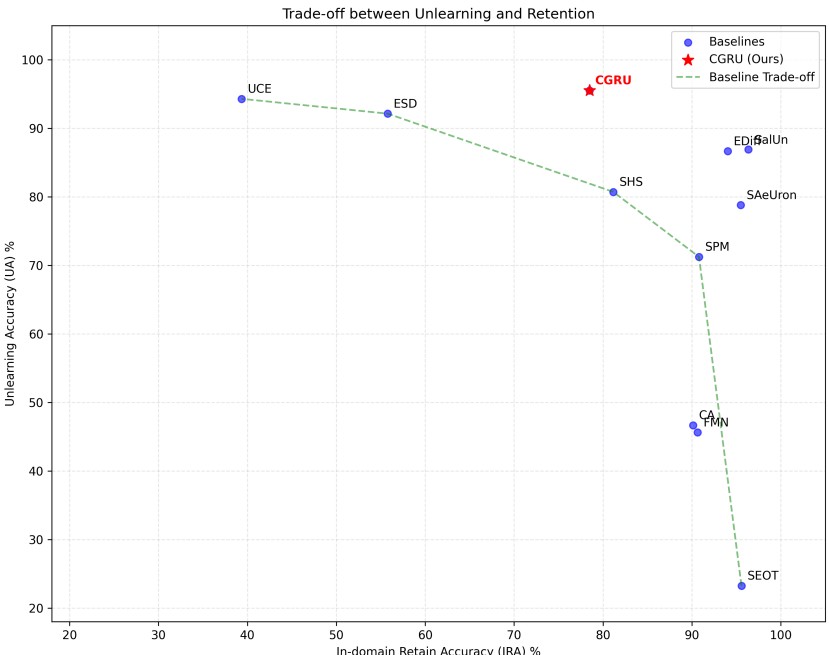

Figure 5: Trade-off between Unlearning Accuracy (UA) and In-domain Retain Accuracy (IRA). The green dashed line indicates the trend of existing baselines. CGRU (red star) lies above this trend, demonstrating a superior trade-off point favoring effective erasure while maintaining competitive retention.

## E.2 GRADIENT VARIANCE ANALYSIS

To empirically validate our theoretical claims regarding stability, we analyzed the variance of the policy gradients during training. Figure 6 compares the mean gradient variance of CGRU against DDPO on the *Aesthetic score* objective. CGRU demonstrates consistently lower and more stable gradient variance throughout the optimization process, confirming the benefit of the per-timestep critic in reducing the high variance associated with sparse terminal rewards.

## E.3 QUALITATIVE PROGRESSION ANALYSIS

To provide deeper insight into the unlearning dynamics, we present qualitative examples showing the progression of concept erasure for three randomly selected classes. The prompts for these results were generated by ChatGPT using the following instruction: "Generate 3 simple prompts for image generation for each of these 20 classes: {list of classes}".

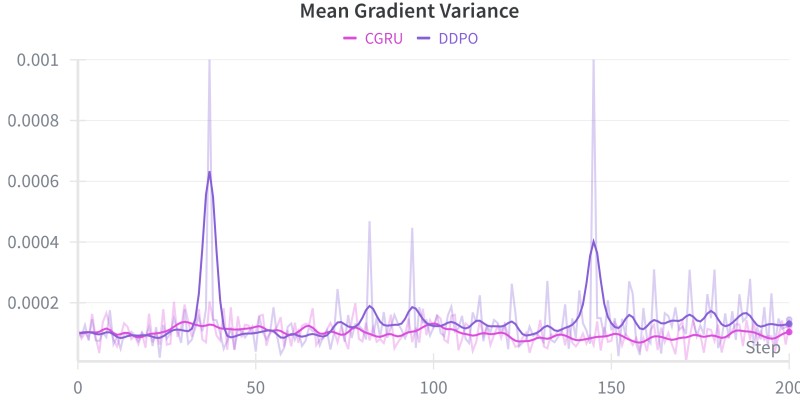

Figure 6: Comparison of mean gradient variance during training for CGRU and DDPO. CGRU exhibits lower and more stable variance, validating the effectiveness of the per-timestep critic for variance reduction.

Figures 7, 8, and 9 illustrate the evolution of generated images throughout the training process. In each figure, the leftmost image corresponds to the untrained Stable Diffusion checkpoint (0 steps). Moving to the right, the images represent generations from checkpoints at increasing training steps. This progression highlights the gradual suppression of the target concept. We also observe that the convergence speed—the rate at which a concept is erased—varies across different classes, likely due to differences in concept frequency and semantic density within the pre-training data.

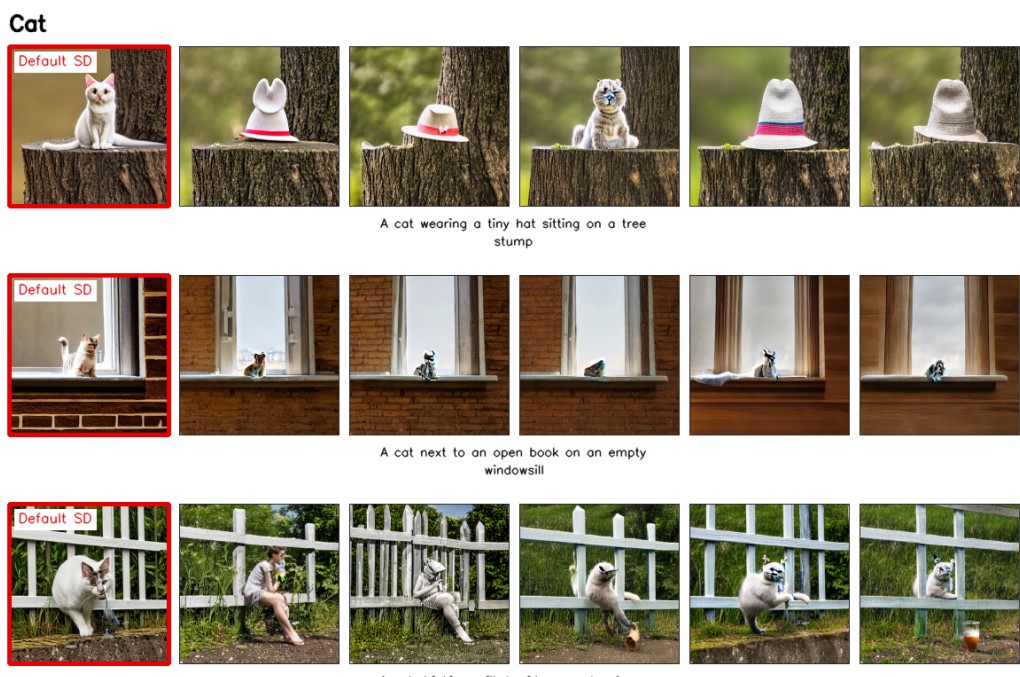

Figure 7: Qualitative progression for the "Cats" class. The first image (left) is the untrained Stable Diffusion checkpoint, and images further to the right correspond to later training stages, showing the gradual erasure of the cat concept.

**Bear**

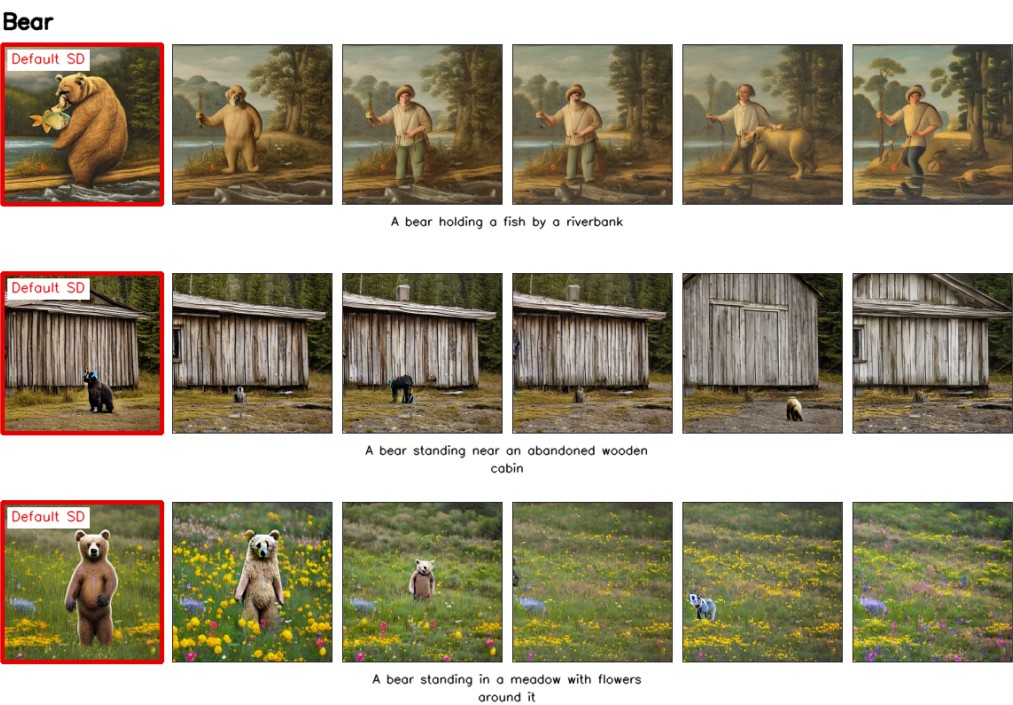

Figure 8: Qualitative progression for the "Bears" class. The leftmost image represents the untrained baseline. As training progresses (moving right), the bear concept is effectively suppressed.

**Dog**

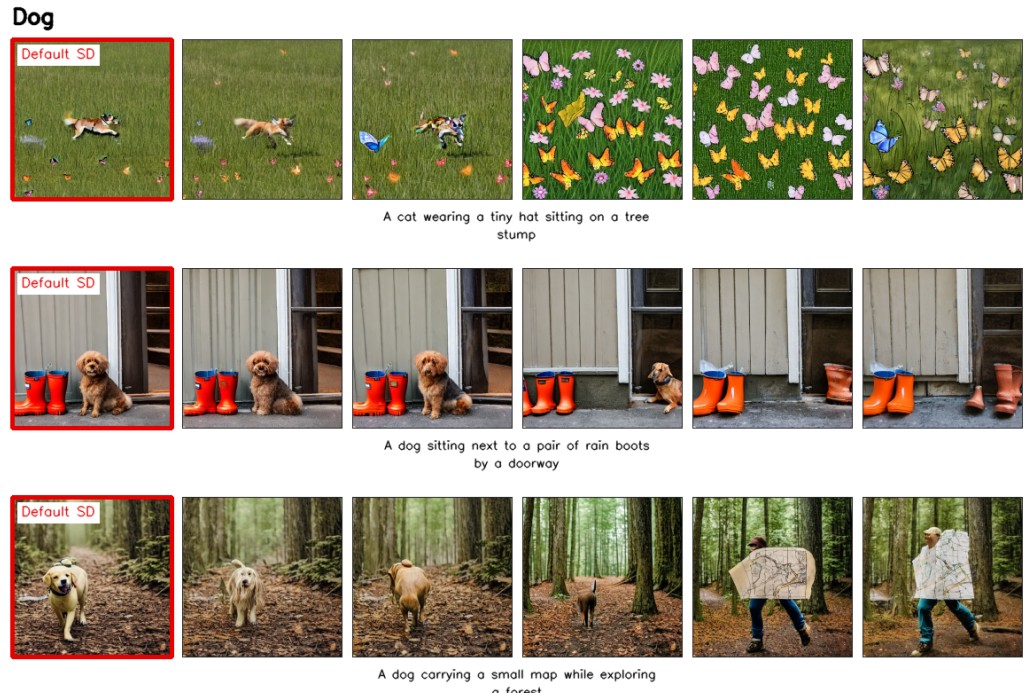

Figure 9: Qualitative progression for the "Dogs" class. Progression from the untrained model (left) through the training process (right), demonstrating the removal of the dog concept.

# F  EVALUATION METRICS

## F.1  FRÉCHET INCEPTION DISTANCE (FID) COMPUTATION

For distribution-level image quality, we report the Fréchet Inception Distance (FID). Specifically:

- **Backbone:** Inception-V3, pool3 activations (2048-D).
- **Computation:** Let $\mu_r, \Sigma_r$ be the mean and covariance of reference features, and $\mu_g, \Sigma_g$ those of generated features. FID is

$$\text{FID}(\mathcal{R}, \mathcal{G}) \;=\; \|\mu_r - \mu_g\|_2^2 \;+\; \text{Tr}\big(\Sigma_r + \Sigma_g - 2(\Sigma_r \Sigma_g)^{1/2}\big).$$

  For numerical stability we add a small diagonal term $(10^{-6}I)$ to covariances prior to the matrix square root.

# G  LARGE LANGUAGE MODEL USAGE

We used Large Language Models (LLMs) to aid and polish the writing of this paper. The LLMs were employed for:

- **Writing assistance**: Helping with sentence structure, clarity, and flow
- **Language polishing**: Improving grammar, style, and academic tone
- **Content organization**: Assisting with logical flow and section transitions
- **Technical writing**: Ensuring consistent terminology and precise mathematical descriptions

All technical content, experimental results, mathematical formulations, and scientific claims remain entirely our own. The LLMs were used solely as writing tools to improve the presentation and readability of our research, following standard academic practices for manuscript preparation.

