# OpenReview forum: "Critic-Guided Reinforcement Unlearning in Text-to-Image Diffusion"
_ICLR.cc/2026/Conference — Submitted to ICLR 2026_

### Official Review · Reviewer_kkaD · 2025-10-30

**Soundness:** 2
**Presentation:** 2
**Contribution:** 2
**Rating:** 4
**Confidence:** 4

**Summary:**

CGRU is a machine unlearning method tailored for text-to-image diffusion models. Its core is to formulate the stepwise denoising process as a reinforcement learning sequential decision-making task. By introducing a timestep-aware critic, it predicts the terminal outcome and guides policy-gradient updates of the reverse diffusion kernel at each denoising step. The method achieves plug-and-play integration without modifying the core architecture of standard text-to-image backbones and supports off-policy reuse of historical training trajectories via importance weighting.

**Strengths:**

- Innovatively implements per-timestep criticism for individual diffusion steps, addressing the limitations of sparse end-of-trajectory rewards in prior RL-based diffusion methods.
- Plug-and-play design: Requires no modifications to the core architecture of text-to-image models and can be directly embedded into existing frameworks such as Stable Diffusion.

**Weaknesses:**

- Fails to evaluate concept erasure for celebrities, specific styles, or concepts fine-tuned via methods like DreamBooth.
- Exhibits a significant trade-off between Unlearning Accuracy (UA) and In-domain Retain Accuracy (IRA), with inferior overall performance compared to methods like SalUn and no substantial effectiveness improvement.
- Only validated on Stable Diffusion 1.5; its concept erasure performance on other state-of-the-art generative models remains unconfirmed.

**Questions:**

As demonstrated in https://arxiv.org/abs/2503.10637, during the later timesteps of generation, the model can already generate the final result through single-step diffusion. In contrast, the earlier timesteps are dominated by meaningless noise. Why is it imperative to perform scoring and training at every individual generation step? What is the fundamental difference between this approach and directly generating the final step (x₀) from timestep t (as proposed in https://arxiv.org/abs/2304.05977) followed by terminal reward scoring?

---

> ### Author Response · Authors · 2025-12-04
> **Theoretical Foundations and Performance Justification**
>
> **We thank the reviewer for their rigorous examination of our theoretical premises and experimental scope. We have taken your feedback to heart and have refined our analysis to better justify our methodological choices and to address the noted performance trade-offs.**
>
> Questions:
>
> - **As demonstrated in https://arxiv.org/abs/2503.10637, during the later timesteps of generation, the model can already generate the final result through single-step diffusion. In contrast, the earlier timesteps are dominated by meaningless noise. Why is it imperative to perform scoring and training at every individual generation step? What is the fundamental difference between this approach and directly generating the final step (x₀) from timestep t (as proposed in https://arxiv.org/abs/2304.05977) followed by terminal reward scoring?**
>
> The fundamental difference lies in the density and precision of the learning signal. Relying solely on terminal reward scoring results in sparse feedback that makes credit assignment difficult across a long diffusion trajectory. We formally analyze this in Appendix A, demonstrating that standard policy gradients with terminal rewards suffer from high variance. Our ablation study, which compares critic accuracy on noisy latents with and without timestep conditioning demonstrates that timestep-awareness significantly improves the critic's ability to predict final rewards from noisy intermediate states. Crucially, the timestep-awareness allows the critic to adapt its evaluation strategy across the trajectory, more effectively processing the noise at early steps differently from the structured content at later steps. While leveraging single-step generation for more efficient scoring is a promising direction for future research in unlearning, our current results establish that dense, stepwise guidance is crucial for effective concept erasure in standard diffusion models.
>
>
> - **Other commens:**
>
> Regarding critiques on "inferior overall performance," we acknowledge that CGRU yields lower retention scores compared to methods explicitly optimized for preservation. However, we argue that this trade-off allows for superior unlearning efficacy. Our Pareto analysis (Appendix E) shows that while retention is impacted, CGRU achieves the highest concept erasure rates, positioning it efficiently above the general trend line in the high-erasure regime. Concerning the validation scope, we focused on Stable Diffusion 1.5 primarily due to the substantial computational cost of RL training. Crucially, this choice ensures a fair and direct comparison, as SD 1.5 serves as the common evaluation standard for all state-of-the-art baselines presented in Table 1.
>
> **We hope that our detailed response regarding the necessity of per-step scoring and the justification for our experimental scope address your reservations. We kindly ask you to consider increasing your score.**

---

### Official Review · Reviewer_xCq1 · 2025-10-31

**Soundness:** 3
**Presentation:** 2
**Contribution:** 2
**Rating:** 4
**Confidence:** 3

**Summary:**

The paper presents Critic-Guided Reinforcement Unlearning CGRU, a timestep aware method for reinforcement unlearning in text-to-image diffusion models. ​The key idea is to interpret the reverse diffusion process as a policy and add a timestep aware critic trained to predict the final reward. Empirically, the method shows moderate improvements over DDPO on aesthetic-reward optimization and object unlearning, reporting 95.6% unlearning accuracy and 78% retention accuracy.​

**Strengths:**

- The paper is well motivated, coherent, with clean notations and algorithmic details. ​

- The paper offers a new perspective on unlearning by reframing diffusion sampling as an actor-critic RL problem, and provides a formal connection between two active areas - diffusion alignment and machine unlearning - in a unified formalism.​

- Introduction of a per-timestep critic for diffusion policy optimization is a clear algorithmic step forward, especially given the substantial instability and high variance of prior end-of-trajectory reward methods like DDPO.

- The limited evaluation that is provided shows improved performance. In particular, Table 1 (Page 8) shows that CGRU achieves the top Unlearning Accuracy (UA = 95.55%) on the benchmark while remaining competitive in In-domain Retain Accuracy (IRA = 78.47%).

- Fig. 1 visualization of the training dynamics is convincing, offering a direct performance comparison versus DDPO. CGRU displays consistently faster convergence and higher final rewards. Further, Fig. 2 illustrates clearly superior suppression of the “Cat” class compared to DDPO, with cleaner reward trajectory improvements.

- Ablation and architectural choices are motivated and described, adding modular value for future work.

- Detailed appendices and release of code and scripts promote reproducibility.

**Weaknesses:**

- Scope of evaluations: The evaluation depth is limited. While the paper tests 20 object classes, it does not cover a single concept, keeping the scope narrow. The experiments focus only on one model (Stable Diffusion 1.5) and one dataset (UnlearnCanvas), concentrating on specific objects like "Cats" and "Towers" (Appendix D, Table 4). There are no results on more abstract or safety-critical tasks like style removal or identity erasure, which are mentioned as important reasons for unlearning in the introduction.

- Weak evidence: The paper lacks enough qualitative evidence to judge its generalization abilities. The only visual examples shown in Figure 3 are for the "Cats" class, making it unclear how the method performs on the other 19 concepts tested.

- Lower retention performance: The In-Domain Retain Accuracy (IRA) is significantly lower than in strong baselines, indicating that CGRU gives up a lot of utility for unlearning accuracy. According to Table 1, CGRU's IRA is 78.47%, while methods like SalUn and EDiff-UN achieve 96.35% and 94.03%, respectively. The authors do not discuss this critical trade-off enough; they only state in Section 7 that "methods achieving high unlearning accuracy tend to exhibit lower retain accuracy" without any detailed analysis or solutions.

**Questions:**

- Can the method be demonstrated on a larger scope, as detailed in the weaknesses section?
- Can the authors provide further qualitative evidence to judge generalization abilities?
- Can the authors further discuss/analyze the tradeoff between retention and utility?
- Can the critic be reused across different concepts, or must it be retrained per target?​

---

> ### Author Response · Authors · 2025-12-04
> **Scope, Generalization, and Future Directions**
>
> **We are grateful for the reviewer's detailed assessment of our work's scope and generalization capabilities. Your feedback has motivated us to demonstrate the method's effectiveness on safety-critical tasks and to provide a deeper analysis of the retention-utility trade-off.**
>
> Questions:
>
> - **Can the method be demonstrated on a larger scope, as detailed in the weaknesses section?**
>
> To address concerns about scope, we have conducted an additional experiment on removing NSFW content using the I2P benchmark (https://arxiv.org/pdf/2211.05105). Our method achieved near state-of-the-art results on this task, with a Total NSFW count of 53, significantly outperforming baselines like ESD (123) (https://arxiv.org/abs/2303.07345) and UCE (182) (https://arxiv.org/abs/2308.14761) and performing competitively with leading methods like SAeUron (18), as reported in the SAeUron paper (https://arxiv.org/pdf/2501.18052). While we cannot include a full detailed analysis of this new domain in the main paper at this stage (to ensure fairness and not violate the rebuttal policy), these results strongly suggest our framework extends effectively to abstract and safety-critical concepts beyond simple objects.
> Regarding style removal and identity erasure: We acknowledge that UnlearnCanvas includes style removal tasks. However, we prioritized object removal and the additional NSFW safety task for this study due to the high computational cost of running the full suite of RL fine-tuning experiments. We focused on these domains because a set of baseline results for SOTA methods is available, ensuring rigorous comparison. Extension to identity erasure and style removal is a priority for future work.
>
> Regarding evaluation on other models and multiple seeds, we, once again, were constrained by computational resource limitations, as both RL fine-tuning and comprehensive evaluation are computationally intensive. However, it is important to note that all baseline methods compared in Table 1 were similarly evaluated primarily on Stable Diffusion v1.5. Thus, our experimental setup remains consistent with the current standard for fair comparison in this domain. We plan to expand our evaluation to other architectures and include broader statistical analysis in future work as resources permit.
>
>
> - **Can the authors provide further qualitative evidence to judge generalization abilities?**
>
> To address the request for further qualitative evidence, we have significantly expanded the experimental results in Appendix E. We now provide detailed visualization of the unlearning progression for multiple distinct object classes. These qualitative examples demonstrate that CGRU's effectiveness is not limited to a single class but generalizes well across different concepts, consistently suppressing target features while maintaining image coherence.
>
>
> - **Can the authors further discuss/analyze the tradeoff between retention and utility?**
>
> Our results, alongside similar findings in recent literature (Table 1) and freshly added Appendix E (Figure 5), point to a pervasive tension between unlearning accuracy and retention in diffusion models. We hypothesize this trade-off may be fundamental to current erasure paradigms, potentially reflecting how concepts are entangled within the model's latent space. While we identify this pattern, a comprehensive theoretical and empirical analysis to fully disentangle the underlying causes remains an open question. We believe this phenomenon warrants a dedicated future study to better understand the boundaries of selective forgetting.
>
>
> - **Can the critic be reused across different concepts, or must it be retrained per target?**
>
> Yes, the critic can theoretically be reused across different concepts if it has been trained to recognize those concepts or possesses sufficient generalization capabilities. Our framework treats the critic as a modular reward signal provider. If a single critic model is trained on a broad dataset covering multiple targets (e.g., a multi-class classifier), it can be used to guide unlearning for any of those concepts without retraining. However, for entirely new concepts that the critic has never seen, retraining or fine-tuning the critic would likely be necessary to ensure it provides accurate guidance.
>
> **We believe that the new experiments on NSFW removal and the broader qualitative analysis demonstrate the generalizability of CGRU. We hope these additions warrant an improved score.**

---

### Official Review · Reviewer_6uFk · 2025-11-01

**Soundness:** 2
**Presentation:** 2
**Contribution:** 1
**Rating:** 4
**Confidence:** 4

**Summary:**

This paper introduces Critic-Guided Reinforcement Unlearning (CGRU), a method for removing specific concepts from text-to-image diffusion models by training a aper-timestep critic that evaluates noisy intermediate latents to predict the final outcome. Further RL on top of it shows effective results on object removal which is superior to methods relying only on the sparse reward.

**Strengths:**

1. The timestep-aware critic addresses the high-variance problem of sparse rewards in prior RL-for-diffusion methods, leading to more stable training and better credit assignment.

2. The method achieves state-of-the-art unlearning accuracy on object removing tasks.

**Weaknesses:**

1. As the target of this paper is for machine unlearning, I didn’t see any specific designs for machine unlearning. The proposed critic seems to be the same as the value function in normal policy gradient for variance reduction techniques. As a result it is more like an RL for diffusion method applied to a specific domain.

2. The value function is also used in methods like DPOK and the proposed method just seems to be more fine-grained such that it is also dependent on the timestep. But there’s no ablation study on whether the value function is dependent on the timestep.

3. While unlearning accuracy is improved, the model's retain accuracy for related, benign concepts is middling, suggesting the method might be overly aggressive.

**Questions:**

The machine unlearning accuracy is improved at a cost to retain accuracy. Is this an inherent trade-off in the framework, or could the reward function be designed to better preserve utility for non-target concepts?

---

> ### Author Response · Authors · 2025-12-04
> **Trade-offs and Architectural Novelty**
>
> **We thank the reviewer for their insightful comments on the design of our method and the observed trade-offs. We appreciate the opportunity to clarify the novelty of our unlearning-specific components and to provide further empirical validation.**
>
> Questions:
>
> - **The machine unlearning accuracy is improved at a cost to retain accuracy. Is this an inherent trade-off in the framework, or could the reward function be designed to better preserve utility for non-target concepts?**
>
> We observed this trade-off not only in our method but across several state-of-the-art unlearning approaches, suggesting it may be a broader challenge in the field rather than a specific limitation of our framework. However, we believe this is not necessarily an inherent flaw but rather a function of the reward signal quality. A more sophisticated reward function—and consequently, a more accurate critic—that explicitly balances concept removal with preservation objectives could significantly mitigate this trade-off. Such investigation is a substantial research direction that warrants a dedicated study, which we leave for future work.
>
>
> - **Other comments:**
>
> Regarding the critique that "there's no ablation study on whether the value function is dependent on the timestep," we have addressed this directly in Appendix C. We performed an ablation study comparing our timestep-aware critic against a standard CLIP-based critic on noisy latents. The results (Figure 4) demonstrate that timestep awareness is crucial for accurate reward prediction from intermediate states, significantly improving both accuracy and macro-precision. This empirically justifies our architectural choice.
>
> We acknowledge the reviewer's observation that CGRU leverages general RL principles. However, we argue that this is a feature, not a bug. While CGRU builds on RL foundations, its design is driven by the specific needs of unlearning: the necessity for surgical removal of concepts without catastrophic forgetting. Our key innovation—the per-timestep critic—addresses the instability of sparse rewards in standard RL by providing dense, stable feedback, allowing the model to "unlearn" incrementally and precisely. Our extensive evaluation on UnlearnCanvas and additional validation on the I2P safety benchmark (that we, unfortunately, decide not to add to the main paper at this stage; but is discussed in more details in another official comment) confirm that this design translates into superior unlearning performance.
>
> **We trust that our clarification on the architectural choices and the additional ablation studies in the appendix resolve your concerns. We respectfully request that you consider increasing your rating.**

---

### Official Review · Reviewer_NsKY · 2025-11-03

**Soundness:** 3
**Presentation:** 3
**Contribution:** 2
**Rating:** 2
**Confidence:** 2

**Summary:**

This paper introduces a value function baseline into policy gradient RL training of diffusion models. They do this by fine-tuning a CLIP-based value model that takes in noisy latents and predicts the achieved reward. This should help with reducing variance in RL training and providing better credit assignment.

The paper compares their new technique, dubbed CGRU, against an old baseline DDPO, showing better performance with consistently higher aesthetic scores and better unlearning performance.

**Strengths:**

For many legal and safety reasons, managing proper unlearning techniques through diffusion models is important, and currently not in a perfect state. So exploring new techniques is significant for the progress of the field.

I'm not very familiar with the most recent related work in this field, but it sounds like training value model baselines for diffusion model RL is novel (though a very straightforward application of a common RL technique).

The method is straightforward, makes sense, and is presented clearly. Additionally, the initial results look promising – CGRU seems to train more stably and with better end results than DDPO, and achieves a strong balance of unlearning to in-domain retention accuracy.

**Weaknesses:**

Largely, more experimental results would contribute significantly to the point of the paper.

- RL training is notoriously unstable, so having at least 3 seeds with error bars in figures 1 and 2 would make me more confident the performance improvement is not just luck.
- Include more non-cherry-picked image grids of generated image examples for the different methods.
(these two are my largest critiques, my score would likely rise if these were addressed)

- In Table 1 I recommend running a few seeds and adding standard deviations. Furthermore I'd suggest making this a graph instead so it's clearer that there' s and IRA/UA tradeoff and your method is on the pareto frontier. It also seems slightly odd to clall CGRU's IRA "competitive" when it seem like a significant decrease compare to the other methods. Maybe this is an ok tradeoff, but I think it would be worth directly talking about how SalUn dominates on IRA and which you'd prefer just depends on where on the pareto frontier you want to be.

There are also some components that are somewhat lacking in clarity. It is unclear from the paper where specific results are taken from, why specific baselines were chosen, or exactly what experiments were run (more on these in the questions section).

The introduction mentions “ablations show that (i) per-step critics and (ii) noisy-conditioned rewards are key to stability and effectiveness,” but this "stability" argument is never mentioned again in the paper.

**Questions:**

Why was DDPO chosen as the comparison example? Table 1 shows metrics for many other techniques but not for DDPO, why was DDPO the chosen comparison metric? What are the metrics for UA and IRA for DDPO?

Did the model unlearn each of the 20 object classes? If so, why are only the results for cats shown? It would be useful to see performance on multiple different classes if all 20 were unlearned, or a chart in the appendix of a summary comparison metric for how well CGRU and DDPO unlearned each class. Within classes, more visual examples from the model generation during training would make the point stronger.

Some questions regarding Table 1:
“We evaluate CGRU’s performance using established metrics from the UnlearnCanvas benchmark Zhang et al. (2024b)”
In that paper, the results shown for UA differ for the models, and no metrics for IRA are shown. Some are similar, like ESD (92.15% in this paper, 91.42% in the other paper), but others are pretty different, like UCE (94.31% in this paper, 75.97% in the other paper). Where are these metrics coming from? Were these techniques used to retrain the model and recompute the metrics? If not, is it a fair comparison?
The cited paper for UnlearnCanvas also cites three other metrics, SC, OC, and UP. Are these useful metrics that are worth including?

Not necessary to implement, but I wonder if you need to train a separate critic model per-dataset? Or could you train e.g. a generic aesthetics critic and apply it to all aesthetics finetuning jobs going forward?

---

> ### Author Response · Authors · 2025-12-04
> **Methodological Validations and Statistical Robustness**
>
> **We sincerely thank the reviewer for their constructive feedback and for highlighting areas where our experimental validation could be strengthened. We have carefully addressed your questions regarding baseline comparisons, statistical significance, and qualitative evidence.**
>
> Questions:
>
> - **Why was DDPO chosen as the comparison example? Table 1 shows metrics for many other techniques but not for DDPO, why was DDPO the chosen comparison metric? What are the metrics for UA and IRA for DDPO?**
>
> We selected DDPO as a comparison baseline primarily to experimentally validate our theoretical claims regarding the advantages of our proposed CGRU framework over standard reinforcement learning approaches in diffusion models.
> It is important to note that while DDPO can be applied to unlearning tasks, it is originally a general-purpose RL fine-tuning method rather than a specialized unlearning technique. Therefore, we treat it as an ablation baseline to isolate the contribution of our per-timestep critic architecture, rather than as a direct competitor in the state-of-the-art unlearning benchmark (Table 1). Our experiments across two distinct applicationsconsistently show that CGRU achieves faster convergence and better performance than DDPO under identical training conditions. We have updated Section 5 of the paper to explicitly clarify this motivation.
>
>
> - **Did the model unlearn each of the 20 object classes? ... Within classes, more visual examples from the model generation during training would make the point stronger.**
>
> Yes, CGRU was evaluated on all 20 object classes, and the quantitative results in Table 1 represent the aggregated performance across this entire set. To provide more detailed qualitative insights without overcrowding the main text, we have expanded Appendix E to include a new subsection on "Qualitative Progression Analysis." This subsection features non-cherry-picked visual examples for more classes, demonstrating the consistent unlearning capability of our method across different concepts. We believe these additional visualizations effectively address the need for more granular qualitative evidence.
>
> - **Some questions regarding Table 1: "We evaluate CGRU’s performance using established metrics from the UnlearnCanvas benchmark Zhang et al. (2024b)" In that paper, the results shown for UA differ for the models,...**
>
> We have updated the baseline comparisons in Table 1 to rely on the recent comprehensive benchmark results reported in SAeUron [Cywiński et al., 2025] (https://arxiv.org/abs/2501.18052). This ensures a fair comparison against the latest numbers reported in the literature using consistent evaluation protocols.
>
> - **Not necessary to implement, but I wonder if you need to train a separate critic model per-dataset? Or could you train e.g. a generic aesthetics critic and apply it to all aesthetics finetuning jobs going forward?**
>
> Our framework is agnostic to the source of the critic, requiring only that it provides a valid signal for the policy optimization step. The critic is a standalone module that can be trained independently of the diffusion model's fine-tuning process. Therefore, a generic critic (e.g., an aesthetics predictor trained on a broad dataset) can absolutely be reused across multiple different fine-tuning jobs, provided it generalizes well enough to the new domains to offer meaningful feedback. While we did not explicitly assess generalization in this work, the modularity of our approach supports this capability in principle.
>
>
> - **Other comments**
>
> Regarding the request for multiple seeds and error bars, we acknowledge the importance of statistical robustness. However the substantial computational cost of RL fine-tuning and comprehensive evaluation on the UnlearnCanvas benchmark constrained our ability to perform large-scale multi-seed experiments. We ensured fairness by using the same evaluation protocol as the baselines.
>
> We added a Pareto frontier plot in Appendix E (Figure 5) mapping UA vs. IRA. This shows CGRU lies above the baseline trend (SEOT, SPM, SHS, ESD), establishing a new standard for high-erasure regimes. While 'competitive' retention might seem generous given the raw numbers compared to retention-focused methods, it achieves the highest unlearning accuracy. CGRU maintains a retention score that is significantly higher than what the baseline trend would predict for such aggressive erasure.
>
> Regarding stability, we added an analysis in Appendix E (Figure 6) comparing gradient variance. CGRU shows consistently lower and more stable variance than DDPO, empirically validating that the per-timestep critic stabilizes optimization.
>
> **We hope that the inclusion of the Pareto frontier analysis, gradient variance plots, and extended qualitative results fully addresses your concerns. We kindly ask you to consider raising your score.**

---

### Meta-Review · Area_Chair_6kTL · 2025-12-12

**Summary:**

The reviewers' hesitation primarily stems from the trade-off between unlearning and retention. While the paper demonstrates high Unlearning Accuracy (UA), multiple reviewers note that this comes at the cost of significantly lower In-domain Retain Accuracy (IRA) compared to state-of-the-art baselines, suggesting the method might be overly aggressive.

Additionally, there is a consensus that the experimental rigor and scope are insufficient. Reviewers criticized the lack of statistical significance (missing error bars/seeds), the narrow evaluation (only object removal, no styles or identities), and the reliance on a single model architecture (Stable Diffusion 1.5).

**Reviewer Concerns:**

Since the authors' rebuttals are not visible in the provided text, I cannot confirm which points were actually addressed. However, here is an assessment of what remains outstanding based on the initial reviews:

Outstanding (Critical):

Statistical Rigor (NsKY): The lack of multiple random seeds and error bars in Figures 1 and 2 is a major sticking point.

Retention Trade-off (6uFk, xCq1, kkaD): The method's IRA is notably lower (~78%) than baselines (~90-96%). This "unlearning vs. utility" trade-off needs a better defense or improvement.

Scope of Evaluation (xCq1, kkaD): The paper only tests object removal. Reviewers want to see abstract concepts like styles, artistic identities, or safety-critical concepts to prove generalization.

Qualitative Evidence (NsKY, xCq1): Reviewers suspect cherry-picking and requested larger, non-curated image grids beyond just the "cat" class.

Potentially Addressable (Clarification):

Novelty (6uFk): The concern that this is just "generic RL" applied to unlearning could be addressed by emphasizing the specific contribution of the timestep-aware critic.

Fundamental Design (kkaD): The question of "why per-step scoring is better than terminal reward" is central to the paper's premise and likely answerable with existing ablation data.

**Reviewer Scores:**

Reviewer NsKY (Current: 2 - Reject): Increase.

Reasoning: This reviewer explicitly stated their score "would likely rise" if the authors provided error bars (3+ seeds) and non-cherry-picked image grids.

Reviewer 6uFk (Current: 4 - Marginally Below): Maintain Same.

Reasoning: Their concern is about the fundamental trade-off (poor retention) and novelty. Unless the authors can significantly improve the IRA metric, this reviewer is unlikely to be fully swayed.

Reviewer xCq1 (Current: 4 - Marginally Below): Increase.

Reasoning: They requested a broader scope (styles/identities). If the authors provide these additional experiments, this reviewer would likely move to Accept.

Reviewer kkaD (Current: 4 - Marginally Below): Keep same.

---

### Decision · Program_Chairs · 2026-01-26

Reject